# A Novel Hybrid Platform for Live/Dead Bacteria Accurate Sorting by On-Chip DEP Device

**DOI:** 10.3390/ijms24087077

**Published:** 2023-04-11

**Authors:** Annarita di Toma, Giuseppe Brunetti, Maria Serena Chiriacò, Francesco Ferrara, Caterina Ciminelli

**Affiliations:** 1Optoelectronics Laboratory, Politecnico di Bari, Via E. Orabona 6, 70125 Bari, Italy; a.ditoma@phd.poliba.it (A.d.T.);; 2CNR NANOTEC—Institute of Nanotechnology, Via per Monteroni, 73100 Lecce, Italy

**Keywords:** di-electrophoresis, *Escherichia coli*, microfluidics, antimicrobial susceptibility testing, chip-scale platform

## Abstract

According to the World Health Organization (WHO) forecasts, Antimicrobial Resistance (AMR) will be the leading cause of death worldwide in the next decades. To prevent this phenomenon, rapid Antimicrobial Susceptibility Testing (AST) techniques are required to drive the selection of the most suitable antibiotic and its dosage. In this context, we propose an on-chip platform, based on a micromixer and a microfluidic channel, combined with a pattern of engineered electrodes to exploit the di-electrophoresis (DEP) effect. The role of the micromixer is to ensure the proper interaction of the antibiotic with the bacteria over a long time (≈1 h), and the DEP-based microfluidic channel enables the efficient sorting of live from dead bacteria. A sorting efficiency of more than 98%, with low power consumption (*V_pp_* = 1 V) and time response of 5 s, within a chip footprint of ≈86 mm^2^, has been calculated, which makes the proposed system very attractive and innovative for efficient and rapid monitoring of the antimicrobial susceptibility at the single-bacterium level in next-generation medicine.

## 1. Introduction

Currently, Antimicrobial Resistance (AMR), i.e., the ability of bacteria to withstand antibiotic treatments, is one of the main issues that modern medicine has to face. According to recent forecasts, the number of deaths due to bacterial infections will increase drastically by 2050 [1,2,3]. This trend is strictly linked to the overuse and misuse of antibiotics. A rapid technique is needed to prevent this phenomenon by enabling the selection of the most appropriate antibiotic. Rapid screening capability of the most promising antibiotic therapy would prove to be strategic in the field of precision/personalized medicine in limiting the spreading of nosocomial infections and, consequently, the onset of antibiotic resistance [4].

The most widely used testing methods include microdilution of bacterial broth and rapid automated instrument methods, based on sensitive optical detection systems allowing the detection of subtle changes in bacterial growth, through materials and devices available on the market. The most well-consolidated approach for AST is microdilution, also called the macro-broth or tube-dilution method, whose advantages include the estimation of the Minimum Inhibitory Concentration (MIC) [5] and the reproducibility of preprepared panels using few reagents. However, the drugs under test are strictly limited by the standard commercial panels’ availability [5].

Enlarging the range of drugs in order to test other methods, such as gradient and disk diffusion, have been proposed in the literature [6,7]. The first method is based on the creation of an antimicrobial agent concentration gradient in an agar medium as a means of evaluating the susceptibility [8,9]. The gradient diffusion method is intrinsically flexible and inexpensive, being able to test any drug at a cost of $2–3/strip [7,10]. This method is the most suitable when one or two drugs are under test or when investigated bacteria require an enriched medium or incubation (e.g., penicillin and ceftriaxone with pneumococci) [7,10]. The disk diffusion susceptibility method consists of placing paper disks saturated with antimicrobial agents on the surface of an agar medium where bacteria are seeded. Incubating the plate overnight is possible to measure the presence or absence of a zone of inhibition around the disks. This technique is well-standardized, due to its simplicity in terms of equipment [11,12]. Although cost-effective, the interpretation of the results is not automated and requires the help of an operator [12]. In short, all the aforementioned methods could provide qualitative assessments, labelling bacteria as susceptible, intermediate, or resistant, but their effectiveness with emerging resistance mechanisms has yet to be demonstrated [13].

An innovative AST is based on the monitoring of bacterial metabolic activity, such as Electrochemical Impedance Spectroscopy (EIS) [14] and impedance flow cytometry [15]. The main drawback of these methods is the very high sample concentration required to achieve a considerable impedance change. To overcome these limitations, photonic approaches have been explored [1,16,17], aiming at performing tests in a very short time (<1 h), but with a small amount of bacteria concentration. As an example, a hybrid platform for trapping and sensing was proposed in [1], where AST is performed via resonance and operating current shift. However, in general, very complex manufacturing and readout techniques are needed, with a lack of portability and system compactness.

Miniaturization of sorting/sensing methods, taking advantage of microfabrication techniques such as optical/electron-beam lithography, micro-milling, or Direct Laser Plotting (DLP), or a combination of these, allows the development of integrated platforms, also useful in liquid biopsy and on-field assays [18,19].

Di-electrophoresis (DEP)-based systems integrated into on-chip platforms could guarantee rapid AST, taking only a few hours, within a very small footprint, exploiting the advantages of both microfluidic and electronic approaches. Since the pioneering work of Pohl [20], the potential of DEP has been investigated for studying and sorting cells, bacteria, viruses, stem cells and proteins [21,22,23]. Efficient sorting of pathogenic and non-pathogenic *Escherichia coli* (EC) has been experimentally demonstrated in [24], using a g-iDEP channel with a pattern of sawtooth-engineered electrodes, driven by an increasing voltage from 0 to 3000 V. A time consumption of 5 s has been demonstrated, at the expense of a large power consumption and footprint (aspect ratio of DEP-based system 41:1).

In this paper, we propose the design of a novel hybrid integrated microfluidic platform for accurate live/dead bacteria sorting. The new device consists of a microfluidic channel combined with a pattern of engineered electrodes to exploit the DEP effect, and a serpentine microchannel where the interaction of drugs with bacteria take place for about 1 h, the time needed to eradicate the bacteria. The engineering of DEP-based electrodes allows the sorting of live from dead bacteria at the outlets, with an efficiency of more than 98% in 5 s, ensuring in this way a rapid and accurate AST in an innovative device. This guarantees accurate and rapid performance of live/dead bacteria sorting, with low power consumption and without labelling and the need for skilled personnel, overcoming the main problems related to the standard counterparts.

## 2. Results and Discussions

### 2.1. Design of the Proposed Platform

The proposed device aims to sort live/dead bacteria through their physical properties. As an example, the metabolic activity of *Escherichia coli* (*ECs*) under antibiotic treatment has been investigated. A three-shell spherical model with a radius *r* of 500 nm (Figure 1a) [25] has been assumed for the live ECs and a one-shell cell (Figure 1b) [26] for dead ECs. It is assumed that the outer membrane can be destroyed when a specific antibiotic is used.

The spherical model has been implemented not only for the manipulation of micro- and nanoscale bioparticles, such as proteins, viruses, algae, bacteria, and blood cells, but also for non-biological particles, such as polystyrene beads, quantum dots, zinc oxide (ZnO) colloids, and silica particles [27]. Even if *E. coli* is shaped like a short rod with rounded ends (sometimes described as ‘pill-shaped’), different studies used the spheroidal model to successfully represent it [25,28]. The operating frequency of the device has been engineered aiming at obtaining opposite signs of the CM factor for the particles under test. The trend in CM factor for both live and dead bacteria is reported in [29], where it can be noticed that, at frequencies < 2 MHz, both living and dead ECs are affected by n-DEP, for those >2 MHz p-DEP can be observed on the live ECs, while the dead are still affected by n-DEP. Therefore, the separation between dead and live bacteria takes place at 2 MHz, which is called crossover frequency. Of course, a variation in the operating frequency leads to a worsening of the device’s efficiency. To preserve all the physical properties of the target bacteria, the Luria Bertani broth (LB broth) has been chosen as the surrounding medium, with a conductivity *σ_f_* = 0.36 S/m. The LB broth is isotonic and isosmotic with ECs [30], allowing it to perform an accurate AMR test.

The properties of the live/dead bacteria and medium at 2 MHz are reported in Table 1. The destruction of the outer membrane leads to remarkable changes in the cytoplasm dielectric constant (108 for live vs. 60 for dead cell) and conductivity (0.22 S/m for live vs. 0.09 S/m for dead cell), which reflect the different CM factors of the live/dead bacteria, as reported in [29].

### 2.2. Design of the Micromixer

The estimation of the antibiotic dose necessary to eradicate bacteria is very challenging. To deal with a bacterial infection, some drugs, i.e., bacteriostatic antibiotics, make the target more vulnerable to the human immune, or temporarily stop the cellular division. However, several bacteria remain in a low metabolic activity state, making the antibiotic treatment ineffective. Only some compounds, such as *beta-lactams*, *ampicillina*, *penicillina*, and *amoxicillina*, target the cellular membrane, causing the death of bacteria [32,33].

Besides the kind of antibiotic, the time-killing kinetics of bacteria are strictly correlated to several factors, such as the bacterium species, the antibiotic concentration and even the culture [32,33]. As an example, kanamycin is able to destroy the cellular membrane of the *EC MG1655* at Minimum Inhibition Concentration (MIC) in LB broth after 40–60 min of exposition, while *ampicillina,* or the bacteriostatic trimethoprim antibiotic, takes around 60–90 min or more, respectively [34,35]. In order to rate the bacteria eradication, the design of the platform has been carried out by considering the *kanamycin* antibiotic, which shows a molecular weight of 600.6 g/mol, a concentration of 23.31 µM, and a diffusion coefficient D_a_ of 5 × 10^−6^ cm^2^/s [34,35]. The MIC of kanamycin for *EC MG1655* (average molecular mass = 10^−12^ g [36], concentration = 0.75 µM, diffusion coefficient D_EC_ = 500 × 10^−6^ µm^2^/s [37]) is 14 µg/mL. In order to enable the proper interaction of bacteria with an antibiotic, a microfluidic micromixer has been designed, consisting of a PDMS serpentine channel. Antibiotics and bacteria should interact within the micromixer for 60 min as a conservative approach to ensure the eradication of *EC* with MIC of *kanamycin* [32,33]. Since the time needed by the particles or molecules to diffuse along the whole channel is equal to ~*d*^2^/*D_i_*, where d is the width of the channel and *D_i_* is the diffusion coefficient of the considered species (*i* = *a*, *EC* for antibiotic or *E.coli*, respectively), the total length of the serpentine channel *L_s,tot_* has been designed to ensure mixing and a time-killing of 60 min. The distance travelled along the serpentine by the flow during this time interval will be *L_s,tot_* = *U_0_ τ*, where *U*_0_ = *Q/d*_2_ is the stripe velocity (cm/s) and *Q* is the flow rate (cm^2^/h). Hence, the minimum length of the serpentine for efficient mixing can be calculated as *L_s,tot_* = *Q/D_i_* [38].

A channel width *d* of 5 μm has been assumed, in compliance with the design of the DEP-based microfluidic channel (see Section 2.2), where *d* has been engineered to optimize the sorting efficiency. The simulations have been carried out through a time-dependent study by using the 2D Finite Element Method (FEM).

To ensure that particles flow in the middle of the DEP channel, *U*_0_ should be equal to 134 μm/s (as demonstrated in [39] and confirmed through simulations) and then *Q* = *U*_0_
*A* = 4.82 cm^2^/h, where *A* is the equivalent area crossed by the flow. To achieve this, the stripe velocities at both inlets should be *U*_0_/2 (77 μm/s). Therefore, since *D_a_ = D_EC_* [34,35,37], the minimum length *L_s,tot_* should be equal to 268 μm to ensure the mixing of bacteria and antibiotic. However, since a time-killing of 60 min should be obtained, the length of the serpentine *L_s,tot_* should be at least 50 cm [34,35]. In order to provide easy fabrication, the serpentine has been designed to achieve an aspect ratio of about 1 *(w_s,tot_/l_s,tot,//_)*, with *l_s__,tot//_*= 9.755 mm and *w_s,tot_*= 8.22 mm (see Figure 2a), preserving the total length that flow has to cross *L_s,tot_.* Therefore, an input length l_*s*,2_ of 2000 μm and a single arm length l*_arm_* of 4000 μm, with outer (*r*_*s*2_) and inner (*r*_*s*1_) radii of 505 μm and 500 μm, respectively, have been designed to preserve the laminar flow of the fluid, with a total number of turns of 50 (Figure 2b). The device dimensions lead to a Reynolds number ≈ 1.65 × 10^−3^ and a Dean number ≈ 5.5 × 10^−4^, so the flow could be considered completely unidirectional [40]. Moreover, to ensure the symmetry of the serpentine channel, the joint structure shows a length (*l*_*s*,1_) of 200 μm, with a bifurcation of 45° with respect to the *x*-axis. All the parameters mentioned are summarized in Table 2 to guarantee a clear overview of the designed structure.

Figure 3a shows the magnitude of the stripe velocity at the micromixer input joint. As expected, the stripe velocity at *l*_*s*,1_ + *l*_*s*,2_ is equal to 134 µm/s, since it is obtained by mixing the flow at the rate of *U*_0_/2. Moreover, mixing between the two species takes place after ~270 µm, as shown in Figure 3b, where the concentration of bacteria and antibiotic *C_a_* (blue curve) and *C_EC_* (red curve), respectively, approach zero at the desired point, with a resulting concentration of 24.06 µM.

### 2.3. Design of the Microfluidic Channel

The design of the microfluidic channel, combined with castellated electrodes, has been carried out by investigating several electrodes’ shaping, squared (Figure 4a), trapezoidal (Figure 4b), and rounded (Figure 4c). All electrodes are arranged as a 1D pattern on both microfluidic channel sidewalls with a period *Λ_i_*, and a width *w_i_*, a height *h_i_*, and a total length *L_to_*_t_ (*i = s,t,r* for squared, trapezoidal, and rounded electrodes, respectively). The width *w_i_* shows a minimum/maximum value *w_t,min_/w_t,max_* for a trapezoidal shape and a value of 2 h for a rounded shape. All the values of those parameters have been proved and optimised through numerical simulations starting from the *w_s_* evaluated by Piacentini, et al. in [39]. Gold electrodes with height and thickness of 20 nm and 200 nm, respectively, were assumed to have applied the needed voltage, as demonstrated in Ref. [40]. All inlets and outlets 200 µm long (*L_in_*) are angled at 45° (*θ*) to achieve the symmetry of the structure and enhance the non-uniform effect of the electric field on cells. The values of *V_P_, L_tot_, d, w_i_, Λ_i_*, have been calculated with a transient analysis by using the 2D FEM approach, aiming at maximizing the sorting efficiency *η_i_*, as defined afterwards, and, at the same time, reducing the power consumption and the operating time. The small thickness of the electrodes (hundreds of nm) allows for simplifying the analysis in a two-dimensional domain, as confirmed by the experimental results reported in [39].

The transient analysis enables tracking of the time-dependent diffusion and convection of particles within the fluid in a laminar flow regime, where the particle motion is mainly dominated by the DEP force and drag forces. Moreover, since the Reynolds number of the particles is much lower than 1, due to the small channel dimension, Stoke’s drag law is used.

Sticky inner wall conditions have been considered in the simulations to take into account the interaction of particles with these inner walls, which results in a decrease in the sorting efficiency.

The device has been designed so that live bacteria, subjected to p-DEP, flow towards the upper outlet, while dead bacteria, affected by n-DEP, flow through the bottom outlet. Inlet #3 is used to pump the LB broth with a stripe velocity of 853 μm/s in order to make the cells flow in the middle of the structure, and to be fully surrounded by the electric field, as reported in [39] and proven by the simulations. This stripe velocity ensures that the injected particles flow in the middle of the channel, where the electric field is large enough to counteract the Stoke force [36].

The width of all electrodes (w_s_, w_t_, w_r_) has been set to 40 µm to maintain the laminar flow through the channel, and the periods (Λ_s_, Λ_t_, Λ_r_) are all equal to 80µm.

The width of the channel (d_s_, d_t_, d_r_) has been assumed to be 5 µm to minimize the power consumption and to ensure the electric field value required for achieving an efficient sorting of all electrode shapes. In addition, to maximize the sorting efficiency and avoid the trapping of the cells within the channel, the length of the microfluidic channel (*L_c,tot_*) has been set to 540 µm. The length has been optimised through simulation and it has been seen that a lower number of electrodes causes a decrement in sorting efficiency, while a higher one provokes the collection of particles near to the electrodes, with a consequent blocking and deterioration of efficiency.

The efficiency *η_i_,* is expressed as the relative percentage ratio of the difference between particles that experience proper and improper sorting, with respect to the total number of cells at the input port:(1)$$\eta_{i}=\frac{\left(N_{live,4}{+N}_{dead,5}\right)\;-\left(N_{live,5}{+N}_{dead,4}\right)}{\left(N_{live}{+N}_{dead}\right)}$$
where *N_live,k_* is the number of the live *ECs* at the outlet *k* (*k* = 4 or 5 for proper and improper sorting, respectively), *N_dead,k_* is the number of the dead *ECs* at the outlet *k* (*k* = 5 or 4 for proper and improper sorting, respectively), and *N_live_* + *N_dead_* represents the total number of cells at the input of DEP-based channel.

Figure 5 reports the efficiency *η_i_* for all the configurations under test as a function of the applied voltage *|V_P_|* at 2 MHz. Maximum efficiency was achieved within the voltage range of 2.3 V–4.2 V, 2.4 V–4 V, and 0.1 V–1 V for squared, trapezoidal and rounded shapes, respectively. It can be noticed that the rounder the electrodes, the lower the power consumption. A rounded configuration ensures the focusing of electrical forces in the middle of the electrode regions, with a consequent need for lower voltage, while, in the squared shape, the force distribution is linear along the electrode region’s cross-section, with a resulting higher power consumption. The trapezoidal shape represents a mid-case scenario.

A voltage *|V_p_|* equal to 0.5 V ensures an average efficiency *η_r_* of 98.75% for the rounded shape electrodes, with an operating time of about 5 s, making the proposed device suitable for rapid antimicrobial susceptibility testing. For the other topologies, a very high-efficiency value has been calculated for a slightly larger voltage value, i.e., *η_t_* = 99.5% with 3.1 V for the trapezoidal, and *η_t_* = 99.5% with 3.2 V for the squared.

During the interaction between antibiotics and bacteria, some bacteria could be damaged only. By assuming a three-shell model with features at halfway between live and dead cells (see Table 1), about 65 % of the damaged cells are collected at the same outlet of the live ECs.

The dynamic DEP response of live/dead cells along the channel is sketched in Figure 6, where the particle flow is captured at different times. The squared shape leads to a slowing down of the flow, caused by an angle mismatch between the electric forces and the particles’ trajectory, with a resulting larger transit time of the channel with respect to other configurations, mainly due to the distribution of the force field, which is much more uniform.

In fact, as highlighted in Figure 7, where the time response for the three topologies of electrodes is reported, the behaviour of the dead *ECs* is the same in all configurations, while live *ECs* are strongly affected by the direction of the electric forces, with a resulting slower time response for the squared shape.

Force orientation in squared electrodes has a very inhomogeneous character, while in rounded and trapezoidal examples field force has the most uniform distribution (see Figure 8). A rounded configuration ensures the lowest time response of about 5 s.

The proposed microfluidic channel could be fabricated by the well-known photolithography and micro-molding technique. The hard master can be produced by SU-8 negative photoresist. Then, the wafer should be exposed to UV light through a film mask, baked, and finally developed by SU-8 developer to remove the unexposed parts. Poly dimethyl siloxane (PDMS) polymer must be poured onto the SU-8 master mold, baked in an oven, and then peeled off. Inlet and outlet holes have to be punched to access the microchannel. Finally, gold electrodes have to be evaporated on both sidewalls of the microfluidic channel to produce a non-uniform electrical field. The electrode pattern can be prepared by electron beam lithography, and the undesired deposed gold is removed through etching techniques.

## 3. Materials and Methods

The hybrid microfluidic platform proposed for AST includes a 1D pattern of gold electrodes aligned on both sidewalls of a microfluidic channel (dotted green box in Figure 9) placed downstream of a serpentine microchannel (dotted red box in Figure 9). The serpentine channel is needed to ensure the proper interaction between drugs (pumped into inlet 2) and bacteria (pumped into inlet 1), for at least 1 h [32,33]. 

The microfluidic channel with integrated castellated electrodes sorts living and dead bacteria in outlets 4 and 5, respectively, by exploiting the DEP physical effect. Inlet 3 is used to pump the fluid.

According to the DEP theory [27], a force (*F_DEP_*) is generated on a dielectric particle when it is surrounded by a non-uniform electric field:(2)$$F_{DEP}{=2\;\pi \varepsilon }_{m}r^{3}{Re[f}_{CM}\left(\omega )\right]\nabla E^{2}\;$$
where *ε_m_* is the absolute permittivity of the surrounding medium, *r* is the particle radius, *Re[f_CM_ (ω)]* is the real part of the Clausius–Mossotti (CM) factor related to the effective polarizability of the particle, *E* is the amplitude (rms) of the electric field, and *∇* represents the gradient operator. The CM factor is always in the range (−0.5, 1) and depends on the structure used to model particles (single-shell model, multi-shell model). In general, at low working frequency, it is mainly affected by particle and medium conductivity values while, at high frequency, it is related to particle and medium permittivity [41,42].

If the polarizability of the particle is larger than that of the fluid (*Re[f_CM_ (ω)] > 0*), the particle experiences a positive DEP (p-DEP) effect, flowing towards the area in which the electric field is more intense, i.e., the force is attractive. On the contrary, if the polarizability of the particle is smaller than that of the surrounding medium (*Re[f_CM_ (ω)] < 0*), the particle is affected by negative DEP (n-DEP), the force is repulsive and the particle moves to where the electric field is weaker [23,24,27,41,42].

As described above, the direction and the strength of the di-electrophoretic force depends on the frequency and the conductivity/permittivity of the particle/medium, which clarifies how live/dead bacteria sorting can be accomplished by exploiting the DEP effect, as well-known in the literature. For instance, in a dead cell, the membrane degrades, with a consequent increase in the membrane conductance and the CM factor. Furthermore, as for the dependence of the force on frequency, it is noted that at low frequencies it depends on the conductivity of the particle and medium, and on permittivity at high ones [43].

The live/dead particle sorting shows higher efficiency when their CM factors show opposite signs, since particles are rejected/attracted (n-DEP/p-DEP) towards positive electrodes. Furthermore, the generated DEP force has to be able to overcome the diffusivity of particles and the drag force (Stokes force) to guarantee the particles’ efficient separation. In the literature, configurations with different DEP magnitudes have been proposed using only p-DEP (*L_channel_* = 3 mm, *h_channel_* = 100 μm, *w_channel_* = 5 mm) [44] or n-DEP (*L_channel_* = 2 cm, *h_channel_* = 25 μm, *w_channel_* = 1 mm) [45], but this approach is less efficient than that of the opposite sign (*L_channel_* = 20 mm, *h_channel_* = 27 μm, *w_channel_* = 3 mm) [46].

To fulfil all microfluidic DEP-based channel constraints, the device sketched in Figure 9 has been designed. Indeed, the serpentine allows bacteria/antibiotic micro-mixing, while the sidewall electrodes on the microfluidic channel generate the proper electric field value that guarantees a trade-off between high sorting efficiency and low power consumption.

The microfluidic device, whose area is approximately 86 mm^2^, is completely made of Polydimethylsiloxane (PDMS), a flexible, optically transparent, low-cost material commonly used in microfluidics due to its favourable properties and fabrication simplicity using soft lithography techniques.

To optimize the DEP forces, the electrode configuration in the microfluidic channel plays a crucial role. The castellated configuration of the electrodes was first introduced by Price, et al. [47], aiming at producing a non-uniform electric field with a high-intensity gradient. This configuration is versatile and very easy to fabricate with standard technological approaches. Several castellated electrode configurations have been proposed in the literature [47,48]. Kurgan, et al. [49] studied the influence of the electrode shapes on the force vectors acting on the particles, including rectangular, triangular, and trapezoidal shapes. For this last geometry, an enhancement of the field force distribution was demonstrated.

In this work, squared, trapezoidal and circular electrode shapes have been studied to demonstrate that, the rounder the shape, the more concentrated the electric field streamlines, following on from the analysis carried out in [49].

## 4. Conclusions

We have demonstrated the feasibility of a high-efficiency process for separating live and dead bacteria in a new hybrid platform, exploiting the di-electrophoresis effect with the aim of contrasting with the AMR. The AMR is a crucial problem, since antibiotics, antivirals, antiparasitic, and antifungal agents become increasingly ineffective owing to resistance developed through their excessive or inappropriate use. A rapid and efficient AST method is needed to properly direct the healthcare staff toward the most powerful antibiotic.

A DEP-based novel hybrid chip-scale platform formed by an interdigitated castellated microfluidic channel, combined with a micromixer, has been designed. A very high sorting efficiency (>98%) has been achieved with a very low value of supply voltage, within a very small footprint, in a very short time. The serpentine channel acts as a micromixer, allowing the mixing of antibiotics and bacteria for a time sufficient to eradicate it (≈1 h). The sorting of live and dead bacteria by means of a castellated-based microfluidic channel enables the evaluation of the efficiency of the drugs on the target bacteria. Several shapings of the castellated electrodes have been investigated: squared, trapezoidal, and circular. The rounded geometry of electrodes allows them to achieve the highest efficiency (>98%), with lower power consumption (*V_pp_* = 1 V), time response of 5 s, and an overall footprint of ≈ 86 mm^2^. This performance overcomes the main drawbacks of the competing solutions, such as high-power consumption, slow time response, and quite high efficiency, making the proposed platform suitable for the next generation of medicines as support to drive the choice of the proper antibiotic in a very short time.

Calculations have been made to sort live/dead *EC*, but obviously the proposed structure can also be used and tested to sort other species of bacteria with the same membrane characteristics as ECs (such as Gram negative *Klebsiella pneumoniae*, *Pseudomonas aeruginosa*, and *Leptospira,* which have demonstrated a change in features when reacting with β-lactam antibiotics [31]) if they have opposite CM factors at the working frequency, guaranteeing high efficiency, rapid separation of species and low power consumption, thereby making a significant contribution to the state-of-the-art. To be effective, the microfluidic channel could be easily adjusted by properly changing |*V_p_*|, while a redesign of the micromixer channel could be needed if the target bacteria needs a time killing larger than 1 h, typical of ECs treated with *kanamycin* antibiotic.

## Figures and Tables

**Figure 1 ijms-24-07077-f001:**
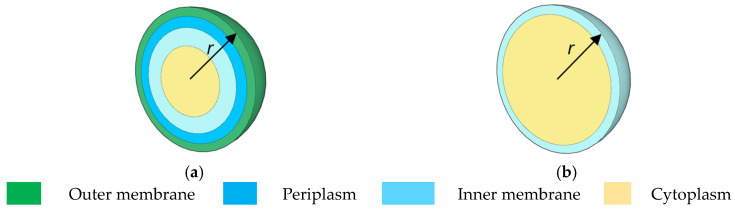
Topological model of spherical (**a**) live and (**b**) dead *E. coli* bacterium [25,26].

**Figure 2 ijms-24-07077-f002:**
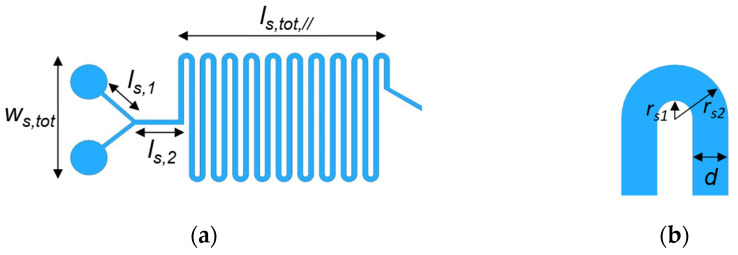
(**a**) Micromixer based on PDMS serpentine channel; (**b**) Zoom of the bent region.

**Figure 3 ijms-24-07077-f003:**
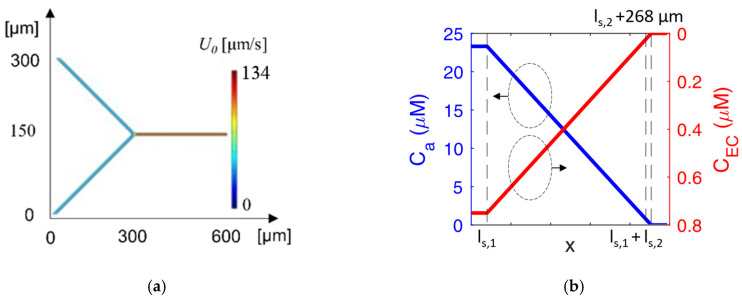
(**a**) Magnitude of the fluid velocity at the micromixer input joint; (**b**) concentration trend of antibiotic (blue curve) and EC (red curve) along the serpentine channel.

**Figure 4 ijms-24-07077-f004:**
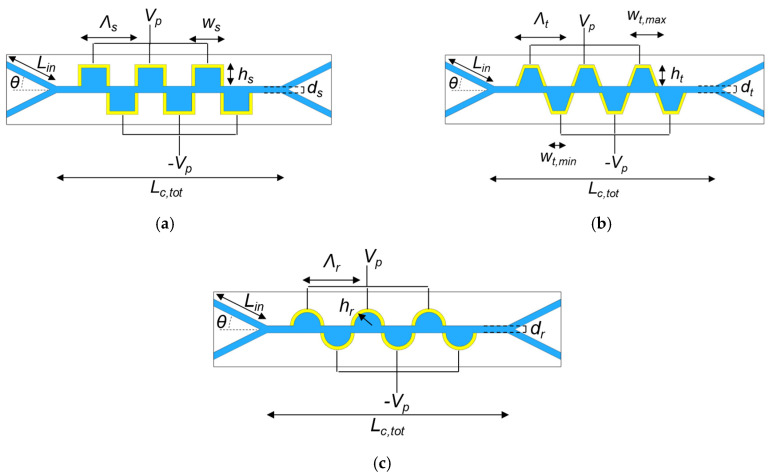
Microfluidic channel (blue) with (**a**) squared, (**b**) trapezoidal, and (**c**) rounded electrodes (yellow) shaping.

**Figure 5 ijms-24-07077-f005:**
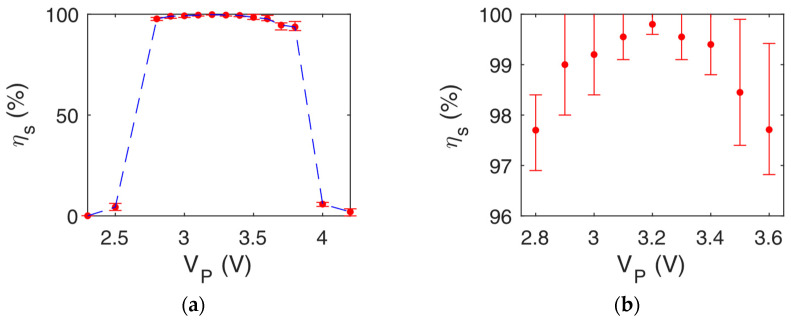
*η_s,t,r_* vs. *V_P_* for (**a**–**b**) squared, (**c**–**d**) trapezoidal, and (**e**–**f**) rounded electrodes shaping for *f* = 2 MHz. The red dots represent the average value after dozens of simulations, with a variance sketched with error bars to take into account stochastic effects.

**Figure 6 ijms-24-07077-f006:**
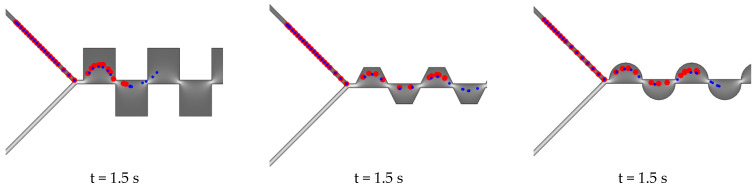
Particle (red: live, blue: dead) sorting for several time steps for squared (*V_P_* = 3.2 V), trapezoidal (*V_P_* = 3.1 V), and rounded (*V_P_* = 0.5 V) configurations: at *f* = 2 MHz. (**a**–**c**) At the beginning of the path, (**d**–**f**) at the middle of the path, and (**g**–**i**) at the end of the path.

**Figure 7 ijms-24-07077-f007:**
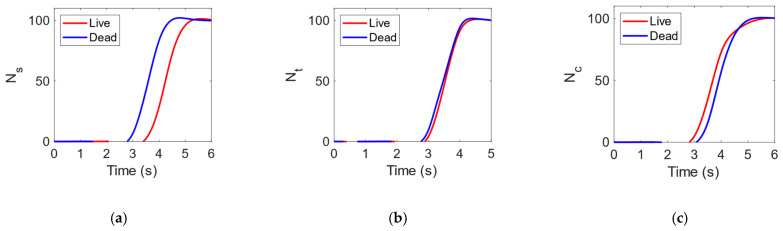
Number (*N*) of live (red) and dead (blue) *E. coli* at outlets #4 and #5. Time response for (**a**) squared (*V_P_* = 3.2 V), (**b**) trapezoidal (*V_P_* = 3.1 V) and (**c**) rounded (*V_P_* = 0.5 V) configurations at *f* = 2 MHz.

**Figure 8 ijms-24-07077-f008:**
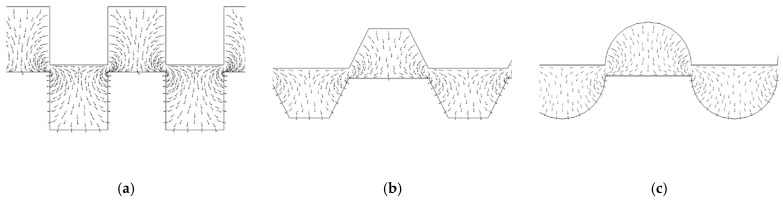
Distribution of the force field in the case of (**a**) squared, (**b**) trapezoidal and (**c**) rounded electrodes.

**Figure 9 ijms-24-07077-f009:**
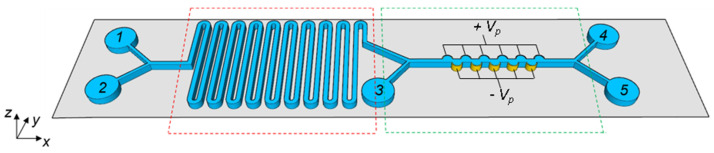
Proposed hybrid chip-scale platform formed by a micromixer channel (dotted red box) and a microfluidic channel with castellated electrodes (dotted green box). Inlets 1 and 2 are used to pump bacteria and antibiotics, respectively, while the outlets harvest the live (4) and dead (5) bacteria. Inlet 3 is used to pump the fluid with a stripe velocity of 853 μm/s.

**Table 1 ijms-24-07077-t001:** Live and dead *E. coli* properties @ 2 MHz [1,26,31].

Property	Live EC		Dead EC	
Density (kg/m^3^)	1116.6	[26]	1184.3	[26]
Outer membrane dielectric constant (a.u.)	10	[31]	-	
Inner membrane dielectric constant (a.u.)	5.5	[31]	5.5	[31]
Periplasm dielectric constant (a.u.)	60	[31]	-	
Cytoplasm dielectric constant (a.u.)	108	[31]	60	[1]
Outer membrane conductivity (S/m)	10^−6^	[31]	-	
Inner membrane conductivity (S/m)	10^−12^	[31]	10^−12^	[31]
Periplasm conductivity (S/m)	3.2	[31]	-	
Cytoplasm conductivity (S/m)	0.22	[31]	0.09	[1]
Radius (μm)	0.5	[31]	0.5	[31]
Outer membrane dielectric constant (a.u.)	10	[31]		
Inner membrane dielectric constant (a.u.)	5.5	[31]		

**Table 2 ijms-24-07077-t002:** Micromixer geometrical parameters.

Parameter	Value [µm]
Channel width (*d*)	5
Total length of mixer (*l_s,tot//_*)	9755
Total width of mixer (*w_s,tot//_*)	8220
Length of inlets (*l*_*s*,1_)	200
Length of straight segment (*l*_*s*,2_)	2000
Length of arm (*l_arm_*)	4000
Inner radius (*r*_*s*1_)	500
Outer radius (*r*_*s*2_)	505

## Data Availability

Not applicable.

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
