# Peer review of "A Novel Hybrid Platform for Live/Dead Bacteria Accurate Sorting by On-Chip DEP Device"

_ijms, 2023, doi:10.3390/ijms24087077_

Round 1
Reviewer 1 Report
The authors developed a hybrid platform for separating live and dead bacteria using the dielectrophoresis effect to combat the issue of antibiotic resistance. The platform consisted of an interdigitated castellated microfluidic channel combined with a micromixer. The castellated electrodes were shaped as squares, trapezoids, and circles, with the rounded shape providing the highest efficiency (>98%) at a low supply voltage and a time response of 5 seconds. The micromixer was shown to allow for the mixing of antibiotics and bacteria to evaluate drug efficacy. The proposed platform was proven to result in high efficiency, rapid separation, and low power consumption, making it a potential solution for the next generation of medicine for quick selection of the proper antibiotic. The platform can be used for sorting different species if they have opposite CM factors at the working frequency.
Overall, the research presented in the article appears to be well-designed and executed. The authors clearly identified a significant problem in the medical field, which is the development of antibiotic resistance and the need for efficient and accurate methods for determining the most appropriate antibiotic treatment. They then developed a new hybrid platform using dielectrophoresis to sort live and dead bacteria, which allows for the evaluation of antibiotic effectiveness.
The authors provided detailed information about the design of the platform and the methodology used in the experiments. The high sorting efficiency (> 98%) achieved with low power consumption and a small footprint is impressive and shows the potential of this technology to be used in the medical field.
One possible limitation of this research is that the experiments were conducted using only one type of bacteria (EC). While the authors mention that the platform could be used to sort different species, further testing is needed to determine its effectiveness with other types of bacteria. Additionally, the use of the micromixer to mix antibiotics and bacteria for a time sufficient to eradicate the bacteria may not accurately reflect the conditions in the human body and further research is needed to determine the effectiveness of this method in clinical settings.
The research presented in the article appears promising and provides a novel solution to a significant problem in the medical field. However, further research is needed to determine its effectiveness with other types of bacteria and its suitability for clinical use. Or a discussion can be added to explain the capacity of the system for other types of bacteria.
How did the authors decide on the electrode and channel geometry?
Reviewer 2 Report
In this paper, the authors mainly propose a novel hybrid integrated microfluidic platform for accurate live/dead bacteria classification.The direction of the dielectrophoretic force is related to the CM factor. If ������������������>0, the dielectric particles will be subjected to positive dielectrophoretic force (pDEP).And the dielectrophoretic force scales with the cube of the particle radius 1. What is the separation principle of live and dead bacteria under the action of electric field?The widespread use of DEP in tumor cell isolation is based on the large difference in size and dielectric constant between erythrocytes and cancer cells. But there is little difference in the relevant data of live/dead E. coli in table1.Authors can add relevant descriptions 2. During the interaction between antibiotics and bacteria, some bacteria are damaged but not completely dead. Which channel will this part of bacteria be assigned to?Author Response
Please see the attachment

Reviewer 3 Report
The article proposes an on-chip platform for rapid antimicrobial susceptibility testing using dielectrophoresis (DEP) effect. The proposed system combines a micromixer and a microfluidic channel with a pattern of engineered electrodes to sort live from dead bacteria. The authors claim that the system has a sorting efficiency of more than 98%, low power consumption, and a time response of 5 seconds, making it suitable for efficient and rapid monitoring of antimicrobial susceptibility at the single-bacterium level. The topic of rapid antimicrobial susceptibility testing is of great importance, given the increasing prevalence of antimicrobial resistance. The proposed system appears to be innovative and promising, as it addresses some of the limitations of existing techniques, such as high power consumption and slow response time.
In general the article is well-written and presents results that are of interst for the readers of IJMS MDPI journal. In order to improve article quality, some points are addressed:
(1) Figure 1, please correct the color in the indication of the cytoplasma that is different in the figure;
(2) At the end of the introductory topic, please highlight to the reader the novelty of the present work in relation to current literature.
